



# Investigation of historical severe storms and storm tides in the German Bight with century reanalysis data

Elke M. I. Meyer[1], Lidia Gaslikova[1]

[1]Helmholtz-Zentrum Hereon, Institute of Coastal Systems, Geesthacht, 21502, Germany

*Correspondence to*: Elke M. I. Meyer (elke.meyer@hereon.de)

## Abstract

Century reanalysis models offer a possibility to investigate extreme events and gain further insights into their impact through

numerical experiments. In this paper, we investigate whether the atmospheric data from the reanalysis models are suitable for simulating historical severe storm tides in the North Sea and whether higher storm tides could have occurred considering different tidal phases. In general, storm tides could be reproduced, and some individual ensemble members are suitable for the reconstruction of respective storm tides. However, the highest observed water level in the German Bight (southern eastern North Sea) could not be simulated with sufficient accuracy. Storms with northerly tracks show less variability in wind speed

and a better agreement with the observed water level for the German Bight. The impact of two severe historical storms that peaked at low tide is investigated with shifted tides, resulting in dangerously high water levels only at Husum in the eastern German Bight.

## 1 Introduction

The German Bight (Fig.1) as part of the south-eastern North Sea is exposed to storm tides, which represent natural hazards for

the low-lying coastal areas. In the last 120 years, a few severe storms occurred with partly considerable damages at the coasts of the German Bight and in the hinterland connected to the North Sea by the rivers (e.g., Ems, Weser, Elbe, Eider). One example is the storm on 16/17 February 1962 which has caused extensive damage due to very high water levels in the German Bight and insufficient protection at the coast and along the rivers. After this event, coastal defences have been significantly improved (e.g., higher dike line and barriers) and the storm tide on 03 January 1976, also one of the highest during the last 120

years, has caused lesser damage mainly due to the improved protection (Kuratorium für Forschung im Küsteningenieurwesen, 1979).

Despite the extensive improvement of coastal protection along the coasts of the German Bight risk of flooding is still present and may increase in the future in the context of anthropogenic climate change. The observational data of water level from the tide gauges extend more than a century back in time and provide an indispensable source of information about extreme storm

events and their impact on the coast. However, these measurements are sporadic by nature and for large parts of the coastline not available. The hydrodynamic models are traditionally used as an additional tool to estimate water levels along the coastline regularly. Storm tides, casing the main flooding hazard at the coasts, can be considered as a composition of atmospherically driven components like storm surge and external surge, tidal component and their non-linear interaction. Leaving aside the changing bathymetry, coastal outline and protection constructions, the main uncertainty in the modelling of historical storm

tides originates in atmospheric forcing. Thus, the realistic reconstruction of wind and pressure fields during the storm is necessary for adequate storm tide estimations. As the observational atmospheric data over the sea are sparse and irregular, especially during the pre-satellite era, atmospheric reanalysis has appeared in the past three decades (e.g., NCEP-NCAR, ERA-40, ERA-Interim, ERA-20C, CERRA). These reconstructions of the atmospheric state are produced with atmospheric models, which also consider available measurements by assimilation procedure.



In recent years, more reanalysis products became available at higher spatial and temporal resolution, improving the representation of localised effects by resolving relevant mesoscale processes (e.g., ERA5 (Hersbach et al., 2020), UERRA-HARMONIE (Ridal et al., 2018, Schimanke et al., 2020)). These reanalyses provide atmospheric conditions for the more recent storms from 1950 onwards. However, events that occurred further back and are still relevant for design purposes are not included. Another type of reanalysis products, which also recently emerged, are the century reanalyses ensembles, e.g.,

Twentieth Century Reanalysis project (20CR), (Compo et al., 2011; Slivinski et al., 2019, 2021). These reanalyses are generated using a weather model, with measurements assimilated. They provide a set of physically consistent atmospheric conditions, which slightly differ due to internal variability in the system. The advantage of the ensemble is that it enables the estimation of uncertainties due to internal variability. Using this as an atmospheric forcing for a hydrodynamic model, it can help to answer the question of what the extreme storm tides would look like if the historical low-pressure fields developed

slightly differently, and thus estimate the uncertainty not only due to variability but also due to imperfectly reconstructed historical conditions.

Another benefit of the 20CR reanalysis is that it goes further back in time, starting from 1836 or even 1806 with an experimental product. Hence, earlier historical storms can be reconstructed than it was possible with other reanalysis products (e.g., see Meyer et al. 2022 for the reconstruction of the 1906 storm). The longer period and multiple realisations of the reanalysis are

somewhat counterbalanced by a coarser resolution (1° or 2°) and even sparser data used for assimilation. The reanalysis products use mainly observed atmospheric pressure data, which are constant in their measurement method and independent in their environment but limited in earlier years and over the ocean. Both factors can influence the storm tide reconstructions driven by these reanalyses.

For the North Sea region in particular, various combinations of atmospheric reanalyses and regional hydrodynamic models

have been used and proved to be valid and effective tools for different water level related studies and applications. For example, Weisse and Plüß (2006) investigated the changes and multi-decadal variability of local extreme water levels using a hydrodynamic hindcast forced by NCEP-NCAR reanalysis refined with the SN-REMO regional model. Later, Weisse et al. (2015) applied a different hydrodynamic model and an improved regionalisation of NCEP-NCAR reanalysis to obtain a 67-year water level hindcast dataset used in several coastal and offshore applications. Arns et al. (2015a) and Arns et al. (2015b)

used yet another hydrodynamic model forced by mean sea level pressure fields, u and v components of the 10 m wind fields of 20CR version 2 reanalysis for 40 years to investigate return water levels and influence of historical sea level rise on storm surge water levels in the German Bight. Vousdoukas et al. (2016) used the ERA-Interim driven hydrodynamic hindcast mainly for validation purposes for the European coasts. While these and other analogous studies nicely demonstrated the applicability of atmospheric reanalysis, they mostly focused on statistics and the long-term evolution of extreme events rather than on the

representation and investigation of multiple separate historical events.

Coastal adaptation measures are usually a long-term and costly effort and a good knowledge of the prevailing environmental conditions and factors contributing to each storm event are important. Among others, some specific studies focused on individual storm events rather than on the statistical properties of extreme storm tides were performed to provide such information. Such studies are, for example, the research projects MUSE, XtremRisk and EXTREMENESS. Therein, either

historical storm tides or storm tides from future climate projections were investigated. In particular, the question of whether the extreme historical storm tides could potentially be exceeded was tackled. In the research project MUSE (Model-based investigations of storm surges with very low probabilities of occurrence on the North Sea coast, Jensen et al., 2006, Bork and Müller-Navarra, 2006) several historical storm tides were investigated. A forecast model was used to create an ensemble of physically possible wind and pressure situations that can cause exceptionally high storm tides. Observed and modelled water

levels were statistically analysed to calculate highest water levels at the North Sea coasts. The project XtremRisk (Oumeraci et al., 2015; Gönnert and Gerkensmeier, 2015) combined results derived from observations of different storm surge components such as storm surge, external surge, tides and their non-linear interactions including future scenarios with sea-



level rise. This project focused mainly on the eastern German Bight and the Elbe estuary in constructing physically possible extreme storm tides. In the project EXTREMENESS, a substantial number of datasets containing reanalyses, hindcasts and

climate change projections were analysed to find highly unlikely but potentially possible storm events with high-risk potential at the southern German Bight. The highest selected events were simulated with different phase lags between the astronomical tides given at the lateral boundaries of the shelf and the wind forcing to analyse if the respective event could become higher (Ganske et al., 2018, Grabemann et al., 2020).

In the present study, we want to further explore historical storm tide events with the help of multiple available atmospheric

reanalyses and address the following questions:

   (a) Are the (century) reanalysis data suitable for simulation of historical strong and severe storm tide events, especially in the earlier period?

   (b) How much variability exists in the atmospheric forcing and how the resulting uncertainty would influence the hazards associated with the extreme storm tides?

(c) What influence does the track position of low-pressure systems have on the water levels in the German Bight?

   (d) Is there a potential for further amplification of extreme historical storm tides, for example, by coincidence with the astronomical high tide?

The paper is structured as follows: in Section 2, there is a brief overview of the historic severe storm tide events in the German

Bight (2.1), and a description of the used reanalysis data (2.2), the hydrodynamic model and tides (2.3). In Section 3 the results are presented and discussed, and Section 4 follows with the conclusions.

## 2. Data and methods

### 2.1 Historical severe storm tide events

The impact of specific storms and storm tides in the North Sea and the German Bight varies along the coastal strips. For example, the storm tide on 03 Jan 1976 affected mainly the eastern parts of the German Bight and the Elbe estuary. The storm tides caused by the 1962 storm belong to the highest events along the entire German North Sea coast and the neighbouring countries. One of the highest observed water levels at the southern German Bight coast occurred during the 13 March 1906 event (van Bebber, 1906 and Deutsches Gewässerkundliches Jahrbuch (DGJ), 2014).

We selected three locations (Norderney, Cuxhaven and Husum) as representatives for the various coastal strips of the German Bight (Figure 1) and listed for each one the three highest water level events that occurred during the past century (DGJ, 2014). Additionally, their classification according to the Federal Maritime and Hydrographic Agency (Bundesamt für Seeschifffahrt und Hydrographie, BSH) is depicted. The classification of storm tide events depends on the average tidal magnitude at the location and exact thresholds for particular locations are listed in Table 1. The BSH defines water levels higher than 1.5m

above mean high water (MHW) as storm tide, a severe storm tide is defined by water levels between 2.5 and 3.5m above mean high water. All water levels higher than 3.5m above MHW are defined as a very severe storm tide. This classification is valid for the Elbe estuary and the eastern German Bight. In the southern German Bight, even lower water levels cause major damage at the coast. In this region, the German DIN 4049-3 (Deutsches Institut für Normung e.V., 1994) classification is also applied. A storm tide is defined as an event, which occurs 10 to 0.5 per year; a severe storm tide 0.5 to 0.05 per year and a very severe

storm tide every twenty years. For Norderney, the water levels estimated for the period 1951-2010 are considered storm tide if they exceed 0.93m, severe storm tide threshold is 2.01m and very severe storm tide is over 2.75 m (Streicher et al., 2015). These values are lower than the thresholds from the BSH definition (Table 1).



| Classification | Definition | Norderney (MHW = 1.14m NAP) | Cuxhaven (MHW =1.46m NAP) | Husum (MHW = 1.58m NAP) |
|---|---|---|---|---|
| storm tide | more than 1.5 m above mean high tide (MHW) | 2.64m | 2.96m | 3.08m |
| severe storm tide | more than 2.5 m above mean high tide | 3.64m | 3.96m | 4.08m |
| very severe storm tide | more than 3.5 m above mean high tide | 4.64m | 4.96m | 5.08m |

Table 1: Storm tide definition by the BSH. MHW is calculated for the period 1961-1990 (https://stormsurge-monitor.eu, 2023); (Normaal Amsterdams Peil, NAP)

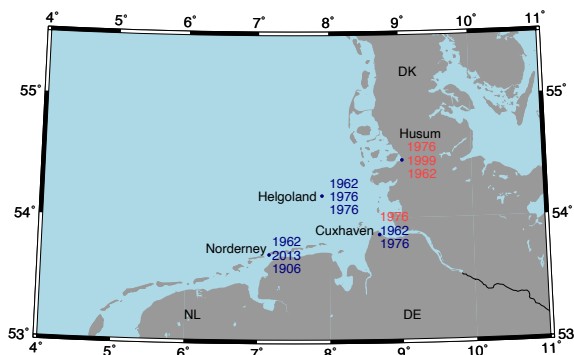

Figure 1: Three highest water level events observed since the beginning of regular observation at selected locations in the German Bight and their classification by the BSH (Table 1) in severe (blue) and very severe (red) storm tides.


The risk potential of individual storms and storm tides changes along the German North Sea coasts depending on the specific storm tracks and associated wind directions and wind set-up. The tracks of storms, which caused the most severe storm tides at one or another substantial region of the German Bight coasts, are shown in Figure 2; their description is complemented in Table 2. The tracks follow the minimum of the low-pressure area for each storm event with the sea-level pressure data derived

from the ERA5 reanalysis starting from 1940. Several storm classification methods according to their tracks were proposed and discussed in the literature in the past decades (e.g., Kruhl, 1978, Gerber et al., 2016 or Prügel, 1941; Schelling, 1952; Petersen and Rhode, 1991). Summarising, the typical storms associated with the storm tide hazard in the German Bight can be separated into the North-West (or Scandinavia) type and the West and South-West (or Jutland) type. In particular, Prügel (1941) defined the types according to the latitude at which they crossed the longitude 8°E (see Figure 2):

-   Jutland type: 55° - 57°N
    -   Skagerrak type: 57° - 60°N
    -   Scandinavia type: 60° - 65°N.

The East Frisian coasts (southern German Bight) have higher water level risks during storms travelling more in the north of Europe. Such storms are labelled as Scandinavia type and are characterized by a long fetch over the North Sea area and high

surge in the entire southern North Sea, e.g., storm tide in 1962 (Rodewald, 1962, Koopmann, 1962 and Gönnert, 2003). The low-pressure system over Scandinavia and at the same time a high-pressure system in the Bay of Biscay generate high-pressure gradients and wind speed over the North Sea (Figure A1, a-c).




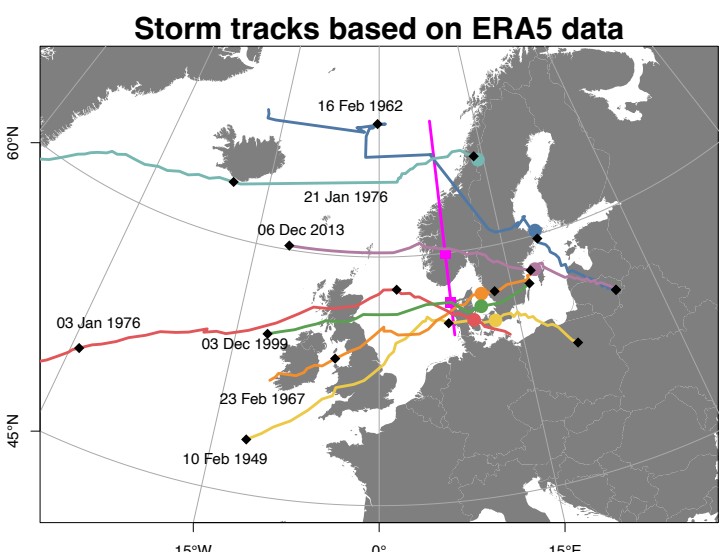

**Figure 2:** Storm tracks derived from ERA5 data. Black diamonds mark midnight on the track. The coloured circles mark a
specific date when the observed water level in Cuxhaven was highest during this event. The magenta line depicts longitude
8°E as the basis for the storm tide classification according to Prügel (1941). The squares separate the types in Jutland,
Skagerrak, and Scandinavia type. For information about mean sea-level pressure during the single events and the Skagerrak
type, see Figure A1.

| Date of event | Type (Kruhl, 1978) | Type (Prügel, 1942) |
|---|---|---|
| 13 Mar 1906 | West and South-West Type | Skagerrak Type |
| 10 Feb 1949 | West and South-West Type | Jutland Type |
| 16 Feb 1962 | North-West Type | Scandinavia Type |
| 23 Feb 1967 | West and South-West Type | Jutland Type |
| 03 Jan 1976 | West and South-West Type | Jutland Type |
| 21 Jan 1976 | North-West Type | Scandinavia Type |
| 03 Dec 1999 | West and South-West Type | Jutland Type |
| 06 Dec 2013 | North-West Type | Scandinavia Type |


**Table 2:** In this study, a selection of severe storms and storm tides in the German Bight is investigated. See also Figure 1.

Fast-moving storms with more southerly tracks, so called Jutland type, are characterized by high wind speeds and steep wind
surge at the eastern German Bight, e.g., the storm on 3rd December 1999. The Elbe estuary is situated between the two regions
and storms of both types were able to cause some of the most extreme observed water levels there. The highest observed storm
tide at Cuxhaven during the storm on 3rd January 1976 was induced by a storm of the Jutland type (Figure A1, d). The situation
was exacerbated by the timing of the tide. Namely, the wind speed peak at the Elbe mouth co-occurred with low tide, thus
preventing the tide propagated earlier upstream the river from being released back to the North Sea, which led to an additional
water level increase on top of the wind surge. The second highest storm tide in the Elbe estuary was caused by a storm of the
Scandinavian type in 1962, which had a prominent impact on the entire German Bight.



Due to partly stochastic nature of maximum storm surge and tidal high water coincidence (e.g. discussion in Horsburgh and Wilson, 2007), some extreme storm surges did not result in extreme storm tides at the coast. Still, such events are of interest because they present the atmospheric conditions, which may lead to extreme water levels when coincide with high tide. So, Figure 2 shows additionally the tracks of two storm events, which caused extreme surges during low tide and therefore did not

lead to very high water levels. The first one, on 10 Feb. 1949 is known for the highest observed surge in Husum and the second one, on 23 Feb. 1967, for the highest observed wind speeds on Helgoland (Tomczak, 1950, Lamb, 2005).

## 2.2 Atmospheric reanalysis products

The basis for this study are the century reanalysis data, in particular the products from the 20[th] Century Project (20CRv2c and 20CRv3) (Compo et al., 2011; Slivinski et al., 2019, 2021). This is an ensemble of global forecast model results with

assimilated observations, e.g., pressure data from the International Surface Pressure Databank (ISPD). Observed pressure data have a long history and are robust concerning systematic measurement errors (Schmidt and von Storch, 1993; Alexandersson et al., 1998, 2000). Thus, the reanalysis dataset represents atmospheric conditions consistent with available observations but slightly varying due to internal variability, especially in the regions with lacking measurement data. In total, 56 members for 20CRv2c and 80 members for 20CRv3 were used. Additionally, to the ensemble reanalyses, the results from the ECMWF

(European Centre for Medium-Range Weather Forecasts) ERA5 and UERRA-HARMONIE, in the following UERRA, reanalysis were used (Hersbach et al., 2018, Ridal et al., 2018, UERRA). These data have higher spatial and temporal resolution, which may have an impact on the quality of the extreme water level simulations. As an additional refinement, the UERRA data were merged with OptempS data (Kristandt et al., 2014). In the OptempS project, high-resolution data in time and space were produced with a German forecast model to get more precise atmospheric conditions for the reconstruction of

storm surges. Forty events starting from 1960 were reanalysed in the project with ERA-40 (Uppala, et al., 2005) and ERA-Interim (Dee et al., 2011) data used as initial conditions (Kristandt et al., 2014). The OptempS data are available approximately three days before and two days after the storm event. The main features of each used dataset are summarised in Table 3.

| Reanalysis | Short form | Number of used ensemble members | Starting year | Spatial resolution | Temporal resolution |
|---|---|---|---|---|---|
| 20[th] Century Reanalysis Project version 2c | 20CRv2c | 56 | 1851 | 2° x 2° | 6-/3-hourly |
| 20[th] Century Reanalysis Project version 3 | 20CRv3 | 80 | 1836 | 1° x 1° | 3-hourly |
| ECMWF ERA5 | ERA5 | 1 | 1940 | 0.25° x 0.25° | 1-hourly |
| ECMWF UERRA-HARMONIE | UERRA | 1 | 1961 | 11 km x 11 km | 1-hourly |

**Table 3:** Reanalysis data by 20CR and ECMWF used as forcing of the tide-surge model.

## 2.3 Tide-surge model

The water level simulations are done with the hydrodynamic tide-surge model TRIM- NP (Kapitza, 2008). TRIM is a Tidal, Residual and Intertidal Mudflat model and was originally developed by Casulli and Cattani (1994) and later nested and

parallelised (-NP) by Kapitza (2008). Pätsch et al. (2017) tested and validated the model, also Gaslikova and Weisse (2013) used the model to simulate multi-decadal hydrodynamic conditions including hindcast and climate change projections (e.g.,



Gaslikova et al., 2013). Callies et al. (2011) used the TRIM-model data for drift simulations. Meyer et al., 2022 used this model to simulate the severe storm tides during the 1906 storm event.

All simulations are done in a barotropic mode within a 3-level nested set-up with regular Cartesian grids, having spatial resolutions from 12.6 km for the North-East Atlantic and the North Sea (grid 1), down to 1.6 km for the German Bight (grid 4), (Figure 3). The tides are calculated separately by FES2004 (Lyard et al., 2006) and introduced at the lateral boundaries of the coarsest grid. All model simulations started at least 15 days before each storm event and used zonal and meridional 10-meter height wind components and sea level pressure fields from the corresponding reanalysis as atmospheric forcing.

To explore the potential for the storm tide amplification for selected events, the historical tide was interchanged with a spring
tide to simulate the highest physically possible water level depending on the respective weather situation during the event.

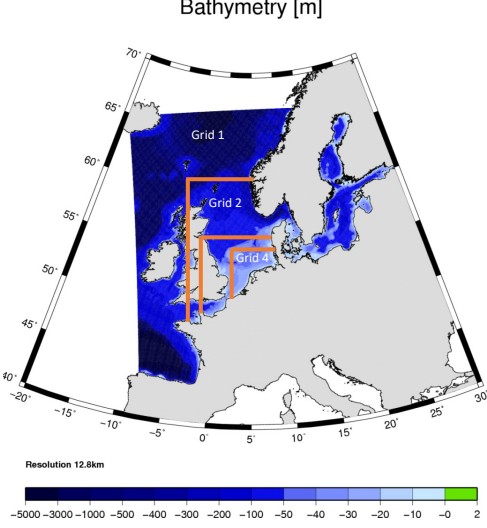

**Figure 3**: Trim model domain with grid 1 (12.8km resolution); grid 2 (6.4km), grid 3 (3.2km) and grid 4 (1.6km)

## 3 Results

On the example of one selected historical storm and a single location in the German Bight, we describe the temporal
development of the storm and exemplarily the relations between different reanalyses and reanalysis members for both the storm tides and the underlying wind conditions. Figure 4 depicts the wind speeds near the Helgoland Island (Fig. 1) during 24 hours before and after the storm peak for the 1962 storm extracted from all used reanalyses as a representative of the marine atmospheric conditions in the German Bight. We do not use the observational wind data for comparison here because the wind measurements for Helgoland are somewhat impaired for certain wind conditions (Lindenberg et al.,2012) and it is not the topic
of the present study to validate the reanalyses as that has been done extensively (Kristandt et al., 2014). During the 1962 storm, the wind speeds in the German Bight were not extremely high, but they exceeded 17.2 ms$^{-1}$ (8Bft, Gale) for a long period. The solid (coloured) lines represent ensemble members of 20CR reanalyses with the largest wind peak maximum. The grey lines show single members with a spread of about 7 ms$^{-1}$ for the peak wind speeds for 20CRv3 (light grey) and about 3 ms$^{-1}$ for 20CRv2c. The OptempS wind speeds, representing here the best guess wind conditions, are on the upper boundary of both
sets of reanalyses. UERRA wind speeds, being very similar to OptempS in the temporal average, exhibit 6-hourly peaks. This is a known feature originating from the UERRA-specific 6-hourly re-forecasting procedure and was discussed by e.g. Schimanke et al. (2020) and Andrée et al. (2021). The peak of the ERA5 wind speed lags behind the peaks of other products and the wind speed is in general lower during the first half of the storm.

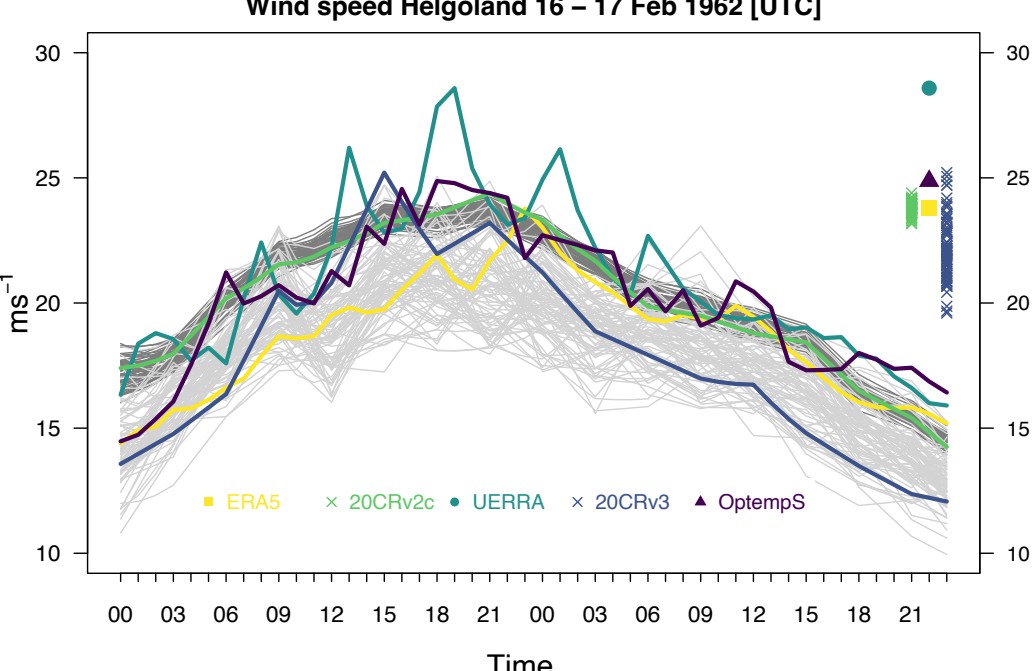

**Figure 4:** Wind speed near Helgoland for the event 16-17 Feb 1962 from UERRA in dark cyan, OptempS in dark violet, ERA5 in yellow, 20CRv2c (all ensemble members – dark grey, the member with the strongest wind – green) and 20CRv3 in light grey and blue for the highest member. On the right side, the highest peaks of the wind speed are shown for each reanalysis and each ensemble member.

The storm tides caused by the described atmospheric conditions are shown in Figure 5 for Cuxhaven, located in the Elbe estuary (Fig.1). Additionally, to the reanalyses, the tide gauge measurements at the location are depicted for comparison. The storm tide event continues for three tidal cycles. For the peak high water the observations lie well within the range of reanalyses ensemble. The range of the peak water levels from different reanalyses and reanalysis members is, however, rather high with about 1.5 m. Looking more into details, it can be inferred that results obtained with UERRA and OptempS forcings are very close to each other and slightly overestimate the observed peak high water. ERA5-driven storm tides underestimate the observed ones, which is in line with the wind speed relation between ERA5 and UERRA/OptempS (Fig. 4). All 20CRv2c ensemble members overestimate observations and most of the other reanalyses, which agrees with the slightly higher 20CRv2c wind speeds during the first part of the storm (Fig. 4). Some of the ensemble members forced by the 20CRv3 are very close to the observations, however the variability of the ensemble set is also large, following the variability of maximum wind speeds.

We will follow the description of storm tide reanalyses for other storm events and locations using only the peak water levels from each realisation for comparison. For notions see an example on the right side of Figure 5.



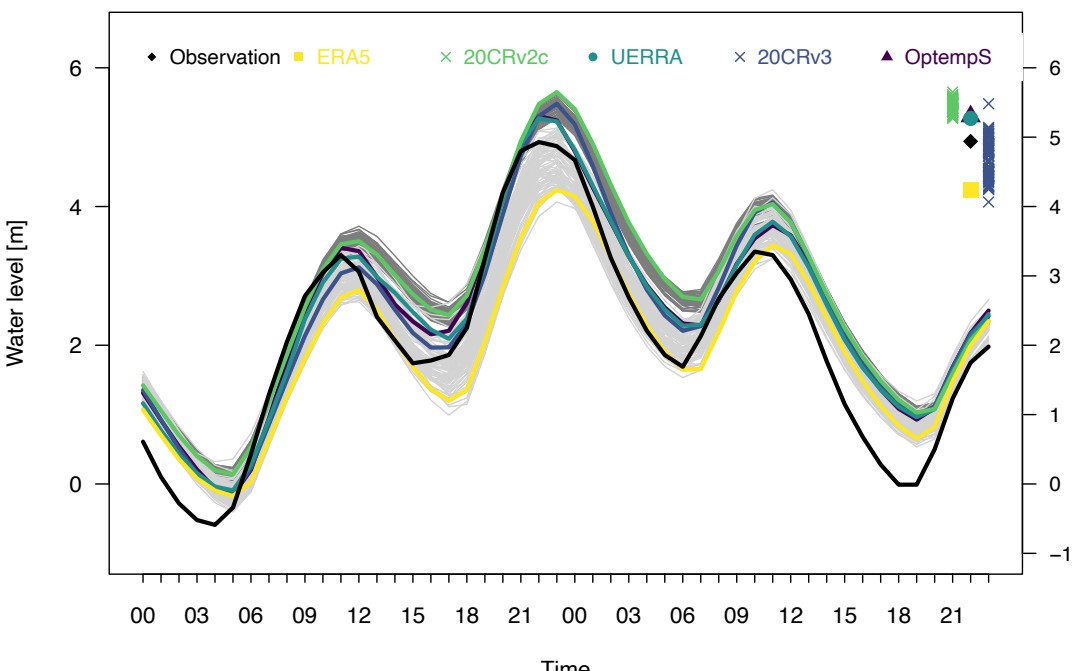

**Figure 5:** Observed and modelled water level for Cuxhaven for the event 16-17 Feb 1962. Observation is in black, simulated results are from model runs forced by various atmospheric reanalyses (see Fig. 4 for colour-coding).


To compare the effects of the eight investigated storms (see Table 2) on the water level at the different coastal stripes of the German Bight, we selected the following stations (Fig. 1): (a) Norderney (southern German Bight), (b) Cuxhaven (Elbe estuary) and (c) Husum (eastern German Bight). The tidal conditions are different for the three selected locations as the tidal wave travels counterclockwise in the North Sea, interacts with the relief and is formed by the combination of dissipation and

reflection. In the German Bight, from east to west and from the open sea to the coast, there are several decimeter differences in the tidal range (Siefert and Lassen, 1996). Therefore, the tidal range at Norderney (2.46 m) is smaller than at Cuxhaven (2.94 m) and Husum (3.50 m), see also (Table A2). This, of course, effects the absolute water level heights during extreme events, but also the relative importance of the surge part for each storm tide.

Figure 6 shows storm tides from modelling results for the selected storm events at the three selected locations together with

the corresponding observed maximum water levels. Additionally, for some events for which the potential for amplification was suspected, the results of additional experiments with tidal shift are shown. They represent the member of the 20CRv3 reconstruction which led to the highest storm tide and for which the water level simulation was repeated with the gradual temporal shift of spring tide instead of the historical tide. For the events with extreme surges that occurred during low tide, the observed low water is additionally depicted.

To discuss the results further, we subdivide the storm events according to the type of atmospheric situation they represent, as was described in Section 2.1. The multi-model ensemble exhibits similar behaviour within each type.

For the events of Scandinavia type, the storms of 1962, 21 Jan 1976 and 2013 with more northerly storm tracks (Fig. 2), the modelling results show, in general, a reasonable agreement with observations (Fig. 6). From the 20CRv3 ensemble, there are



at least several members, which can reproduce historical hydrodynamic conditions for each event and at all three locations.

The 20CRv2c ensemble displays less variability in peak water levels, although at least some members are still close to the observations, except for the 2013 event at Norderney and Cuxhaven (southern German Bight). This is consistent with a smaller variability of wind maxima for this ensemble compared to the next version 20CRv3 (see Fig. 4 and Fig. 8). UERRA and OptempS reconstructions also provide realistic atmospheric conditions for the simulation of extreme storm tides, resulting in estimations of water level maxima lying within 0.3 m (for 1962 and 21 Jan 1976 events) and 0.5 m (for 2013 event) range

relative to observed values for all locations. Simulations driven by ERA5 atmospheric data generally underestimate the observed storm tides at all locations by several decimetres. This result could be anticipated from the wind conditions directly, with ERA5 winds typically having lower extreme wind speeds than e.g. UERRA or 20CRv3 during the selected storms (see e.g. Fig. 4 and Fig. 8). The Scandinavia type of storms often lead to large-scale storm surges, which effect the entire German Bight (e.g. Fig. 7a) leading to severe storm tides in all three selected locations and especially in the southern North Sea. In

particular, during the storm of 1962, although water levels are slightly overestimated by UERRA and 20CRv2c reconstructions, the observed extreme high water levels could be reproduced by at least four common members of the 20CRv3 ensemble with the 10 cm accuracy at all three locations. This suggests the possibility of using the most fitting ensemble members for the reconstruction of the hydrodynamic conditions during this event for the entire coastline of the German Bight, covering with certain confidence even the coastal regions where the observational data were not available.




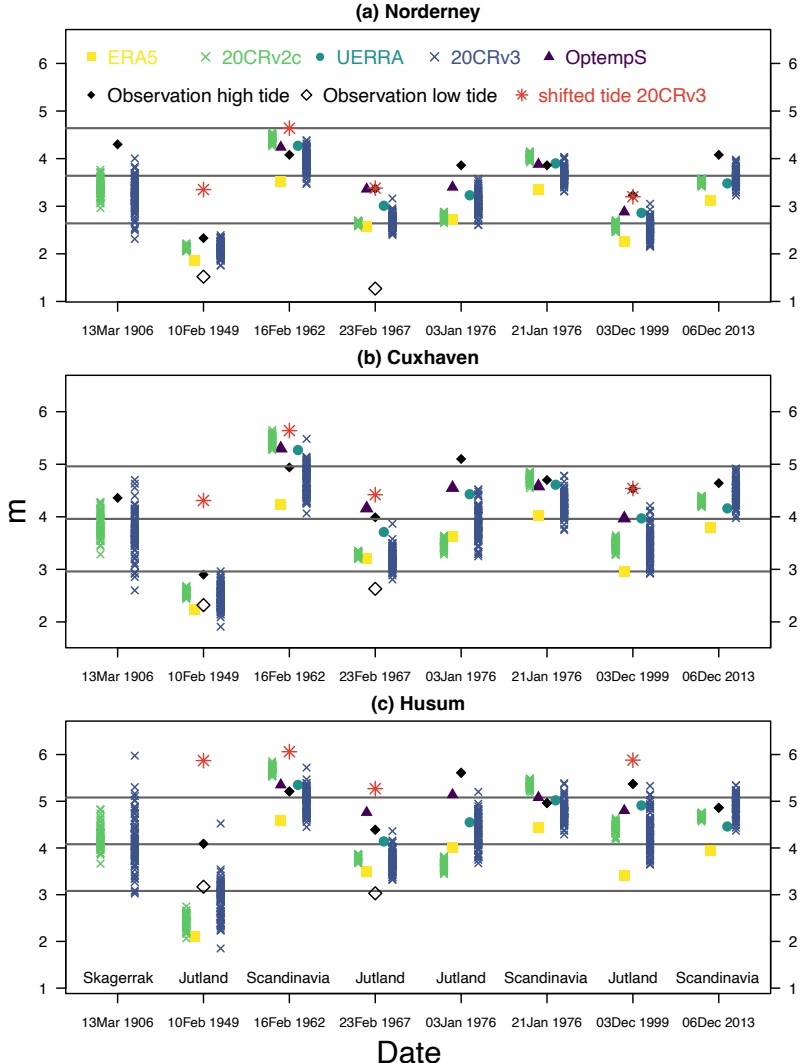

**Figure 6**: Maximum water levels in meter above normal Amsterdam level (Normaal Amsterdams Peil, NAP) for the three selected locations and the eight storm events. The different symbols and colours represent the different atmospheric forcing (see Table 2 and Fig. 4). The black diamonds stand for the observed water levels during high tide (filled) and low tide (unfilled). A red star marks the maximum water level from the tidal shift experiment for a selected member of the 20CRv3 reanalysis. The grey horizontal lines mark the level of very severe storm tide (top), severe storm tide (middle) and storm tide (bottom) for the respective locations (Table 1).

Following the classification of storm tracks, the next type is Skagerrak (Figure A1) with storms moving along more south-located trajectories. One of the representatives of such storms was the event of 13 May 1906, responsible for one of the highest observed water levels in the southern German Bight. The detailed analysis of this storm event, issues with the reconstruction of atmospheric conditions and the quality of the simulated storm tides can be found in Meyer et al., 2022. Here we show, for the sake of consistency, the ability of 20CR reanalysis to reproduce the storm consequences for Cuxhaven and Norderney. For



Norderney, there were three reliable sources for observations available with the maximum water levels ranging from 3.84 to 4.30 m (Meyer et al., 2022). Depending on which observation source is considered more trustworthy, the peak water level is either slightly underestimated by all results of hydrodynamic modelling or selected ensemble members are able to reproduce

the peak water level. For Cuxhaven, several members of hydrodynamic reanalysis show peak water levels close to the observed conditions.

The representatives of the Jutland type of storms – 3 Jan 1976 and 3 Dec 1999 – are relatively fast storms moving through the North Sea almost from west to east (Fig. 2). With the cold front passing through the German Bight, they are characterised by shorter in time but more intensive wind speed extremes from south-westerly/westerly directions and rapid directional changes

shortly before the storm peak in the German Bight (Fig. A4). This makes it more difficult for atmospheric reconstructions to capture and reproduce the particularities of the storm, which is reflected also in the results for peak water levels. All considered reconstructions underestimate the peak water levels for both storms and all locations (Fig. 6). Although some members of the 20CRv3 reconstruction led to water levels close to the observed ones, the previous version 20CRv2c and ERA5 have difficulties to reproduce the exact historical storm conditions. It is assumed that, particularly for the storm on 3 Jan 1976, the

atmospheric reconstructions may be lacking some short-term or small-scale meteorological phenomena crucial for the extreme storm surge formation.

To illustrate the effects of storm tracks on the spatial distribution of water levels in the North Sea, Figure 7 summarises the results for two storm events that exemplify the Scandinavian (16/17 February 1962, Figure 7a-c) and the Jutland type (3 January 1976, Figure 7d-f) of storm tracks. Both events caused the two highest observed water levels in Cuxhaven during the

last century.

The panels show water level (Fig. 7a, 7d), tide (Fig. 7b, 7e) and surge as non-tidal residual (Fig. 7c, 7f) at the time of peak water level in Cuxhaven during the corresponding event as modelled using the merged atmospheric UERRA-OptempS forcing. The tidal component is estimated by model simulations without atmospheric forcing. Non-tidal residuals comprise wind surge and external surge but also consider inverse barometer effect and non-linear tide-surge interactions. In both situations, storm

peaks approximately co-occurred with high tide in the German Bight (Fig. 7b, e). For the event in 1962, the whole German Bight was affected by very high water and surge levels (Fig, 7a, c), whereas, during the event in 1976, only the east coast of the German Bight and the Elbe estuary experienced a very severe storm tide (Fig. 7d, f). Storm tracks of the Jutland type are moving much faster over the North Sea and have a shorter impact on the water level than the storms with northerly tracks. During the storm in 1962, additionally, a large contribution of external surge (Koopmann, 1962) raised the high surge in the

entire southern North Sea. This phenomenon is mainly associated with the influence of low-pressure systems in the North-East Atlantic via inverse barometer effect, then enhancement at the continental shelf and finally propagation into the North Sea (e.g., Böhme et al., 2023). Thus, the ability of atmospheric reconstructions to represent the intensity, location and speed of the low-pressure system in the North-East Atlantic also contributes to a more realistic representation of the storm tides in the German Bight.






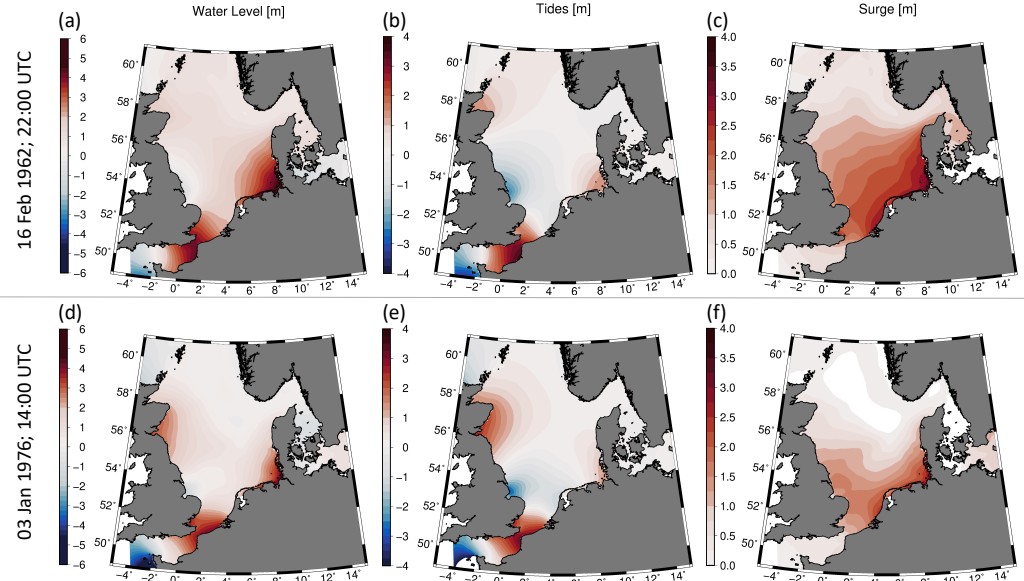

Figure 7: Water level (a, d), tides (b, e) and surge (external and wind surge; c, f) for the event 16 Feb 1962 (upper row) and 3 Jan 1976 (lower row) calculated with wind forcing by OptempS.

Taking advantage of the ensemble reconstructions, we look now at the effect of internal variability for each investigated storm

event (Table 2). As a representative of atmospheric conditions in the German Bight, the wind speeds near Helgoland were selected. Figure 8 shows the maximum wind speeds for each considered storm obtained from each reanalysis or reanalysis member. There are several features of inter-reanalysis relations specific to different storms. For example, OptempS and UERRA wind maxima, showing always one of highest wind speeds among all reanalysis, are sometimes close to each other (e.g. for 3 Jan 1976 or 3 Dec 1999 storms) and for other events are different and exceeding all other reanalysis (e.g. 1962,

1967, 21 Jan 1976 storms). This can partly be attributed to somewhat artificial 6-hourly wind speed peaks in UERRA discussed for the 1962 storm example. These do not have a strong effect on the storm tide extremes, as can be seen for the 1962 and 21 Jan 1976 example storms in Figs. 5 and 6. Another distinction between the storms is the magnitude of 20CRv2c extremes relative to 20CRv3 extremes. Namely, they are sometimes at the upper range (e.g., for 1962), low range (e.g., 1967) or somewhere in between as for all other storms. These relations are not always transferable to the storm tide extremes, e.g., for

the storm of 3 Jan 1976.



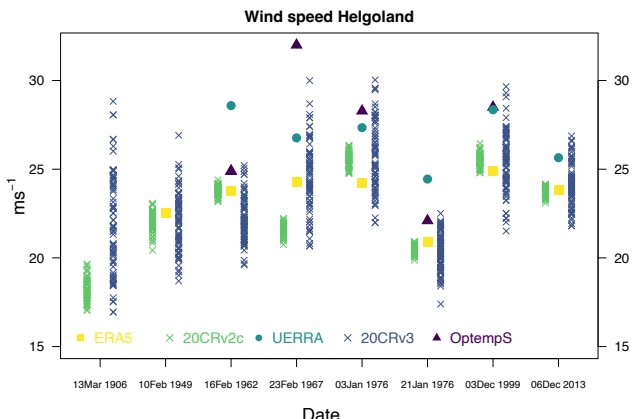

**Figure 8:** Maximum wind speed simulated by the reanalyses for the location Helgoland during each event. See also Table A3 and Figure A4.

There are also some features, which are common for all considered storm events. Such, the wind speed maxima from ERA5 reanalysis are generally lower than those from UERRA/OptempS and are within the range of the 20CRv3 ensemble. It is noticeable that the wind speeds from ERA5 are not higher than 25 ms$^{-1}$ during these severe storm events. Haakenstad et al. (2021) and Dullaart et al. (2020) show, that ERA5 is underestimating very high wind speed compared to the observations. This is also reflected in the related storm tide extremes (Fig. 6), where extreme water levels forced by ERA5 lie at the lower range

of other reanalysis and typically underestimate observations.

   When we look at the spread of the wind maxima for different storms, the earlier storms, in particular 1906, show larger variability than more recent storms. This can be attributed to a smaller number of available observations for that period and thus less assimilated data, which leads to more degrees of freedom for the atmospheric circulation. In the year 1949, a large amount of data from the relevant regions had already been assimilated into the model for the Northern Hemisphere. Only

surface pressure and wind data were used in the 20CR reanalysis, skipping the substantial increase in data available from satellites starting from 1980[th].

   This may increase the uncertainty of the 20CR reanalysis for the recent decades with respect to other datasets, however, it ensures the consistency of the data quality throughout the whole reanalysis period. This, in turn, enables us to notice the following differences in variability between the considered storms: it can be inferred (e.g. Fig. 8 and Table A3) that both 20CR

ensembles demonstrate a larger uncertainty range of the maximum wind speed for the storms of Jutland type and notably smaller uncertainty for the Scandinavia type. As has been described earlier, during the Jutland-type storms the low-pressure area travels directly through the North Sea (e.g., Fig. A1) and the exact position and travel velocity of the low strongly influences the local wind speed and direction over the North Sea. Thus, the minor variations in the storm track or timing due to natural variability led to relatively large changes in the local wind speed maxima in the German Bight. Whereas for the

Scandinavia type storms, with low-pressure areas travelling mostly beyond the North Sea, the wind is not that sensitive to the minor variations in the position and travel velocity of the low. It is also easier for other reanalysis with only one realisation to be more realistic in wind representation for the Scandinavia type, which makes it easier to reconstruct the storm tides more realistic for these types of storms and it takes more effort to make it for the Jutland type (as is the case for the 3.01.1976 storm). Another characteristic common for all storm events is a significantly smaller variability of the 20CRv2c ensemble with respect

to 20CRv3 (Fig. 8 and Table A3). Whereas 20CRv2c wind speed maxima from different members are spread by 1.1 to 2.6 ms$^{-1}$ for a single location depending on the storm event, the 20CRv3 members are spread by 5.1 up to 11.9 ms$^{-1}$. The difference



between 20CRv2c and 20CRv3 originates mainly in the different assimilation schemes and improved forecast systems by the National Centers for Environmental Prediction (NCEP) with higher resolution in both time and space (Slivinsky et al., 2019). The different uncertainty rates are also noticeable in the storm tide ensembles. For 20CRv2c, the peak storm tides spread

between 0.15 m and 1m for various storm events with a median of 0.34 m. For 20CRv3, the spread increases to 0.95 - 2.1 m with a median of 1.16 m. These uncertainties are associated with the natural variability only, e.g., the slightly shifted location, timing or strength of atmospheric low-pressure system and consequent variability in high wind speed directions, duration and magnitude. Being physically consistent with the historical large-scale atmospheric conditions, these atmospheric realisations represent possible realistic developments of certain historic storms and thus realistic storm tides. It should be noted, that some

ensemble members of both 20CR reanalysis lead to higher water levels than observed ones, hinting at potentially possible amplification of storm tides within the historical settings. This should be considered e.g., coastal protection design, which rests upon historic water level extreme among other criteria.

The storm events in 1949 and 1967, mentioned in Table 2 but not discussed so far, exhibited extreme atmospheric conditions

but did not lead to extreme water levels at the coast. The event in 1967 was distinguished by exceptionally high wind speeds in the German Bight, the event in 1949 led to the highest recorded storm surge near Husum (Figures 6 and 8). During both events, the peak water levels were not noticeably high. The observed low waters at Husum, however, were the highest from the beginning of the record, indicating the presence of exceptionally high storm surges. Both events are reasonably represented, although slightly underestimated by at least some members of the water level reconstructions.

These two storm events represent examples of events which possibly could generate more severe peak water levels in case of a more unfortunate temporal coincidence of the storm peaks and high tide, especially spring high tide. To analyse the potential of these historical storm tides for amplification, physically consistent with the real conditions, additional numerical experiments were done. The member of the 20CRv3 ensemble which produced the highest storm tides during each event was selected to investigate whether these storm events could have caused much higher water levels at the coast under different

tidal conditions. In the numerical experiments a spring tide and temporal shift of the tide were used together with the historical atmospheric conditions. It can be concluded, based on BSH categorisation for Norderney and Cuxhaven, that such storms would not result in very severe storm tides in any case, though the 1949 storm had more potential for an increase (about 1m for Norderney and about 1.35m for Cuxhaven). For Husum, if the event of 1949 would have happened under more unfavourable conditions, it would lead to very severe storm tides comparable to the historically observed maximum water

level (Fig. 6). Such considerable amplification can be partly explained by larger tidal range near Husum and the funnel-shaped coastline, which exacerbate the changes by a switch from low tide to high spring tide. Additionally, being the Jutland type of storms, the 1949 and 1967 events caused a more pronounced surge at the eastern coast of the German Bight, effecting Husum at most of the selected three locations.

The possibility of higher water levels during different tidal conditions was also investigated for the events in 1962 and 1999.

The ensemble members with the highest water level for 1962 and 1999 produced by 20CRv3 were shifted to a spring tide. In 1962, the simulated water level was already higher than the observed ones. This meets for all locations and presents the highest water level for all events: 4.64 m for Norderney; 5.64 m for Cuxhaven and 6.06 m for Husum.

The water levels for the 1999 event were underestimated compared to the observed ones, but with the amplification of the tides, the water levels become higher, particularly in Husum.


**Conclusion**

We investigated some of the most prominent storm tides observed in the German Bight during the last 120 years. The water levels associated with the storms were simulated with a tide-surge model using atmospheric forcing from different reanalysis products (20CR, ERA5 and UERRA). The resulting extreme storm tides were compared with observations for three locations



at different coastal strips of the German Bight. The comparison of storm tide extremes with measurements gives a hint on the quality of the wind and pressure data and their capability to represent the atmospheric conditions during extreme storm events. In our investigation, we could show that the historical severe storm tides could be simulated realistically with individual members of 20CRv3, UERRA-Harmonie as well as the merged UERRA-HARMONIE-OptempS reanalysis. Only the 03 Jan 1976 event could not be simulated satisfactorily using any of the considered reanalysis products. The ERA5 data did not

provide higher wind speed than 25 ms$^{-1}$ close to the Helgoland location and all water level results are lower than the observed ones at the coast. Some differences between observed and modelled water levels were expected due to a range of factors not related to the atmospheric drivers. Such, the considered historical storm events occurred within the last 120 years, during which there were significant changes in bathymetry, especially in the shallower areas and estuaries. Changes in the coastline due to erosion, consolidation and extensive protection constructions also took place. Additionally, the mean sea level rise is

manifested in the region with changes of about 25cm during the period. All these changes were not accounted for in the present study. Additionally, there is a limitation due to the hydrodynamic model spatial resolution of 1.6 km and the ambiguity of some observational data, especially for older storms (e.g., 1906).

The maximum wind speeds during the storms showed more variability for the 20CRv3 ensemble than for 20CRv2c and the range encompassed the maxima winds from most of the other reconstructions. It translates also into the variability of storm

tide peaks, leading to the differences from 0.95 m to 2.1 m between the 20CRv3 reconstruction members depending on the storm event. This uncertainty, representing the internal variability of the atmospheric system, indicates the realistic range of extreme storm tides associated with certain atmospheric situations. We have also considered an amplification mechanism due to a combination of atmospheric and non-atmospheric conditions; in particular, we have shifted the tidal high water to coincide with the wind speed peak to get information on what is physically possible for the worst high water level at the coast. In these

experiments, the extreme water levels would increase by a few decimetres. For the storm events 1949 and 1967 with shifted tides, the peak water levels would not be higher than the already observed ones in Norderney and Cuxhaven. For Husum, this experiment produced one of the highest observed or simulated water levels. This can be attributed mainly to the funnel-shaped coastline and larger tidal range near the location but also storm surge distribution, which for these particular storms effected more the eastern than the southern parts of the German Bight. Shifting the tides to spring tide for the 1962 event resulted in

the highest simulated water levels for all three locations. Generally, the shift of the tides shows a higher effect for the eastern German Bight than for the southern German Bight for the considered storms (Figure 6).

Furthermore, in our investigations we have distinguished between Scandinavia, Skagerrak, and Jutland type of storm tracks over the North Sea and investigated their impact on the water levels at the German Bight coast. The northerly storm tracks cause a high surge over the entire southern North Sea. The southern storm tracks cause high surges on the eastern side of the

German Bight (Figure 7). Generally, the water level in the German Bight can be simulated using reanalysis data, but the accuracy of reproducing the observed extreme water levels depends partly on the type of storm track. The simulated water levels in the German Bight coast are more uncertain and mostly underestimated compared to the observations when the storm is categorised as Jutland type. One reason may be the incomplete reconstruction of the fast-running low over the southern North Sea (Figure 2) in the forcing atmospheric data.

In summary, the various atmospheric reanalysis products are useful forcing data for investigating historical storm tides and their effects on the coasts. The study of historical storm tides, on the other hand, may be considered for risk management in case of coastal protection, which rests upon historical water level extremes among others.

However, there are still historical events that would benefit from further improvement of the atmospheric data with new digitised historical pressure data, e.g., the severe storm in the Baltic Sea region in November 1872 (Feuchter et al., 2013;

Rosenhagen and Bork, 2009). Hawkins et al. (2023), for example, have shown that with new digitised historical pressure data and an improved data assimilation process, the reanalysis of the severe windstorm over England and Wales in February 1903 could be significantly improved. Also, the simulation of the severe storm from 03 Jan 1976 needs further improvements. These



examples show that severe windstorms and resulting storm tides need to be studied more closely to avoid coastal flooding and
its consequences.


**Appendix A**

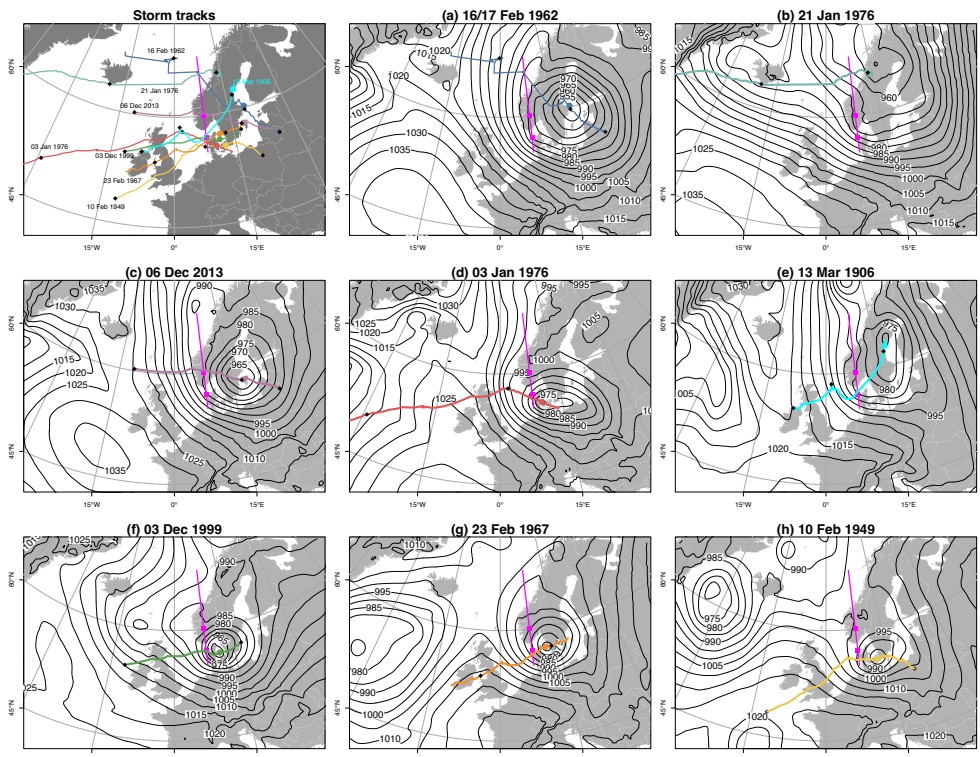

**Figure A1: S**torm tracks and mean sea-level pressure at the high peak of the storm are shown for the Scandinavia type (a-c),
Skagerrak type (e) and the Jutland type (d,f-h). The events are sorted according to the latitude of the storm tracks at longitude
0°. All tracks and mean sea-level pressure data are from ERA 5, only Figure (e) is calculated from 20CRv3.


|  | Norderney | Cuxhaven | Husum |
|---|---|---|---|
|  | Long: 7.2°     Lat:53.7° | Long: 8.7°     Lat:53.9° | Long: 9.0°     Lat: 54.5° |
| mean low water (MLW) | -1.23 m | -1.42 m | -1.80 m |
| mean high water (MHW) | 1.23 m | 1.52 m | 1.70 m |
| tidal range | 2.46 m | 2.94 m | 3.50 m |

**Table A2:** Information about the locations of the tide gauge and their range, 10 years mean (2004-2013). (Deutsches
Gewässerkundliches Jahrbuch, 2014)




| | 20CRv2c | | | | 20CRv3 | | | | OptempS |
|---|---|---|---|---|---|---|---|---|---|
| | Mini-mum | Median | Maxi-mum | STD | Mini-mum | Median | Maxi-mum | STD | Maximum |
| 13 Mar 1906 | 17.04 | 18.21 | 19.65 | 0.64 | 16.94 | 21.79 | 28.83 | 2.84 | -- |
| 10 Feb 1949 | 20.43 | 22.13 | 23.08 | 0.57 | 18.7 | 22.14 | 26.91 | 1.65 | -- |
| 16 Feb 1962 | 23.17 | 23.81 | 24.4 | 0.26 | 19.6 | 22.12 | 25.21 | 1.24 | 24.87 |
| 23 Feb 1967 | 20.75 | 21.6 | 22.23 | 0.32 | 20.67 | 24.52 | 30 | 1.94 | 32.00 |
| 03 Jan 1976 | 24.75 | 25.56 | 26.34 | 0.40 | 21.98 | 25.31 | 30.03 | 1.92 | 28.44 |
| 21 Jan 1976 | 19.87 | 20.48 | 20.92 | 0.23 | 17.4 | 20.26 | 22.5 | 1.08 | 22.21 |
| 03 Dec 1999 | 24.79 | 25.51 | 26.45 | 0.37 | 21.51 | 25.5 | 29.66 | 1.67 | 28.51 |
| 06 Dec 2013 | 23.1 | 23.74 | 24.16 | 0.25 | 21.81 | 24.03 | 26.87 | 1.23 | -- |

**Table A3:** Statistics of maximum wind speed of the ensembles during the severe storm event for the 20CRv2c and 20CRv3 products and OptempS for the location Helgoland (ms⁻¹).


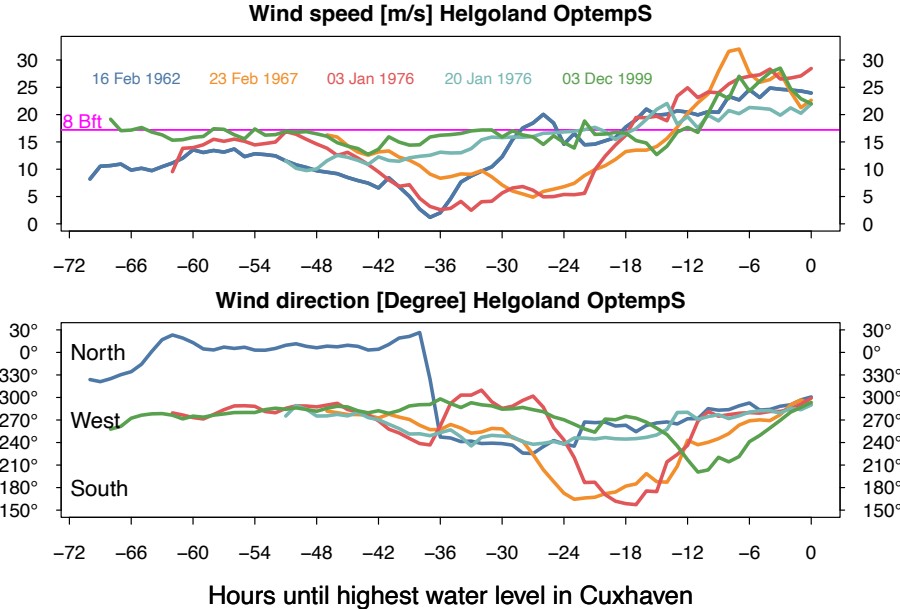

Hours until highest water level in Cuxhaven

**Figure A4:** Wind speed and direction for the location Helgoland in hours before the highest water level in Cuxhaven. The colours indicate the specific event. The light and dark blue colours indicate storms, which are categorised as Scandinavia type

and the others are the Jutland type (orange, red, green).

### Data availability

Meyer, E.: Reconstruction of the 1906 Storm Tide in the German Bright using TRIM-NP, FES2004, and NOAA-CIRES-DOE Twentieth Century Reanalysis (20CR) version 2c and 3, World Data Center for Climate (WDCC) at DKRZ [data

set], https://doi.org/10.26050/WDCC/storm_tide_1906_20CR, 2021.

Meyer, Elke M. I.: *coastDat historical severe storm tides in the German Bight*. World Data Center for Climate (WDCC) at DKRZ. https://doi.org/10.26050/WDCC/CoastdatStormTides. 2023



**Video supplement**

The animation of the reconstruction of the storm tide from 1906 is available on the TIB AV-Portal at
https://doi.org/10.5446/49529 (Meyer et al., 2020).

**Author contribution**
EM did the simulations and analysis. EM and LG have written the manuscript.

**Competing interests**
The authors declare that they have no conflict of interest.

**Acknowledgements**
"Support for the Twentieth Century Reanalysis Project dataset is provided by the U.S. Department of Energy, Office of Science
Biological and Environmental Research (BER) program, by the National Oceanic and Atmospheric Administration Climate
Program Office, and by the NOAA Physical Sciences Laboratory."
UERRA https://doi.org/10.24381/cds.dd7c6d66
ERA5 https://www.ecmwf.int/en/forecasts/dataset/ecmwf-reanalysis-v5
The authors would like to thank Iris Grabemann and Ralf Weisse for fruitful discussions.

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
