# Peer review of "Investigation of historical severe storms and storm tides in the German Bight with century reanalysis data"

_EGUsphere, 2023_

## Author Comment (AC1)

*We would like to thank the two reviewers for taking the time to review our manuscript and for their valuable comments and suggestions to improve our manuscript. We have written our responses below their points in blue.*

*RC1:*
The modelling of storm tide events is highly important for understanding extreme sea levels and coastal flooding, especially from a climatic perspective (past and future). Moreover, because of its complex dynamics, long-term modelling and analysis are challenging. The paper presents the analysis of different reanalysis forcing for storm tide modelling in the North Sea. The paper is well written and well structured, and results are clearly presented, with figures and tables being all relevant.

I compliment the authors for their huge amount of modelling work. I suggest the author use such a database of model results for performing a deeper analysis and discussion on the uncertainty in reproducing extreme events using ensembles (20CRv2c and 20CRv3).

We have extended the discussion of uncertainties in the results section and expanded the appendix with the statistics of water level extremes for both ensembles. The main differences between 20CRv2c and 20CRv3 are the increased temporal and spatial resolution and changed data assimilation scheme for the 20CRv3 ensemble. This results in the more realistic representation of extreme water levels forced by 20CRv3 for the low-pressure systems, which are moving over the central North Sea. More realistic in this case means that the ensemble values are more centred around the observed one rather than overestimate the observations as in the case of 20CRv2. The model results for southern tracks show a higher variability in extreme water levels compared to northern tracks and in general purer accuracy. Still, the 20CRv3 ensemble demonstrates significant improvements with respect to previous version.

From line 382 we will complement the paragraph with:

Another point worth mentioning is that the ensemble mean values of maxima water levels underestimate the observed values almost for all events and both ensembles, with the exception of 1962 and 21 Jan 1976 storms simulated with 20CRv2 (Table A5). Whereas the selected ensemble members can reproduce or come close to the observed extreme water levels, especially for the Scandinavia type storms, the ensemble mean underestimation reaches up to 1.6m. Specifically for Jutland type events, the use of ensemble mean values for the representation of water level extremes is not recommended.

This table is added to the Appendix.

| Cuxhaven | 20CRv2c | | | | 20CRv3 | | | |
|---|---|---|---|---|---|---|---|---|
| | Max - Obs. | Mean - Obs. | Max - Min | STD | Max - Obs. | Mean - Obs. | Max - Min | STD |
| 16 Feb 1962 | 0.71 | 0.52 | 0.38 | 0.09 | 0.54 | -0.26 | 1.41 | 0.28 |
| 21 Jan 1976 | 0.16 | 0.02 | 0.31 | 0.07 | 0.08 | -0.48 | 1.03 | 0.20 |
| 06 Dec 2013 | -0.24 | -0.35 | 0.21 | 0.05 | 0.28 | -0.17 | 0.95 | 0.20 |
| 03 Jan 1976 | -1.46 | -1.63 | 0.36 | 0.09 | -0.58 | -1.23 | 1.27 | 0.30 |
| 03 Dec 1999 | -0.88 | -1.07 | 0.38 | 0.09 | -0.32 | -1.08 | 1.29 | 0.30 |
| 23 Feb 1967 | -0.64 | -0.70 | 0.15 | 0.04 | -0.12 | -0.79 | 1.06 | 0.18 |

**Tab A5:** Statistics of modelled water levels by 20CRv2c and by 20CRv3 and observation (Obs.) for the location Cuxhaven

---

## Author Comment (AC2)

*We would like to thank the two reviewers for taking the time to review our manuscript and for their valuable comments and suggestions to improve our manuscript. We have written our responses below their points in blue.*

RC2

This is a case-study type paper, examining what would happen to flood risk if particular types of historical storms occurred coincidentally with higher tide levels, on the German Bight coastline. It's not particularly novel scientifically, but these studies can be useful as evidence building for local impacts work so I would recommend publishing after some minor revisions. It summarises model runs of storm surges forced by a variety of met reanalyses. It appears to be sound in terms of method and results but doesn't communicate specific messages very well.

The abstract is very bland. Obviously reanalysis can be useful, obviously total water levels depend on tidal phase. I suggest the authors make clearer, and more positive statements - which reanalysis models? How much higher? How much accuracy?

Thank you for the suggested issues, which require more clarification. We reformulated the abstract.

Century reanalysis models offer a possibility to investigate extreme events and gain further insights into their impact through numerical experiments. This paper is a comprehensive summary of historical hazardous storm tides in the German Bight (southern North Sea) with the aim to compare and evaluate the potential of different century reanalyses data to be used for the reconstruction of extreme water levels. The analysis is done based on the results of the regional hydrodynamic model simulations forced by atmospheric century reanalysis data, e.g., 20CR ensembles, ERA5 and UERRA-HARMONIE. The selected eight historical storms lead either to highest storm tide extremes for at least one of three locations around the German Bight, or to extreme storm surge events during low tide. In general, extreme storm tides could be reproduced and some individual ensemble members are suitable for reconstruction of respective storm tides. However, the highest observed water level in the German Bight could not be simulated with any considered forcing. The particular weather situations with corresponding storm tracks are analysed to better understand their different impact on the peak storm tides, their variability and predictability. Storms with more northerly tracks generally show less variability in wind speed and a better agreement with the observed extreme water levels for the German Bight. The impact of two severe historical storms that peaked at low tide is investigated with shifted tides. For Husum in the eastern German Bight this results in substantial increase of the peak water levels reaching historical maximum.

What is new about the experiments with different tidal phase - is the result any different from just adding the difference of tide-only height?

Linear combination of surge and tide, although a useful and widely applied tool for the upper limit estimates, often lead to an overestimation of total water levels due to unaccounted non-linear effects, especially in the shallow-water areas. In the present study, we were looking for maximum possible but still physically plausible storm tides and therefore used modelling experiments. This way we are more confident in our conclusion that for selected storm events the maximum water levels would be significantly higher if the storm would coincide with the different tidal phase. To our knowledge, this is the first time one looked at the systematic shifts of tidal phase for these particular events.

Who should read this paper, and why?

Those responsible for short-term coastal protection measures based on the forecast information and more broadly interested in protection and risk assessment. The take-home messages would be that storms with more southerly tracks are less predictable – here the variability in forecast is often larger and the resulting extreme storm tides may be underestimated; and storms with more northerly tracks may potentially cause longer high water events, which becomes additionally relevant, for example, for the hinterland drainage.

Those interested in the applicability of various atmospheric reconstructions and ensemble simulations for further simulation of extreme events. What is typically used and evaluated when the ensembles are considered, is an average of the ensemble members and thus the extremes are smoothed. Extremes in the ensemble outputs (e.g. near-surface wind speed) or their implications for further processes (e.g. surges and waves) are usually not assessed or considered as higher percentiles of the ensemble range, not particular events. Here we cover the gap and investigate whether at least some ensemble members can represent the historical extremes adequately.

The paper would be easier to read with more headings, for example at line 383 the new section needs a subheading. Elsewhere as well, consider what the reader is supposed to learn from the information presented.

Thank you for the hint. We added more subheadings in the result section.

Line 209:       3.1 Analysis of the storm tide event of 16 Feb 1962
Line 245:       3.2 Other storm tide events in the German Bight
Line 262:       3.2.1 Scandinavia type
Line 287:       3.2.2 Skagerrak type
Line 297:       3.2.3 Jutland type
Line 307:       3.2.4 Differences in water level and surge between Scandinavia and Jutland type
Line 329:       3.3 Variability and uncertainties dure to atmospheric conditions
Line 383:       3.4 Amplification of storm tides by shifting tides

Eg fig 7 - the top and bottom panels are pretty similar events, so what are we looking for?

The top and bottom panels show the water level and surges associated with the two representative storms of different types discussed in the paper. Distinct low-pressure system tracks and speed of the storm passage lead to either local extreme water levels in the parts of German Bight (typical for Jutland type, Fig.7d) or to more large-scale high water level

events (typically for Scandinavian type, Fig.7a). The later also causes higher water levels during several consecutive tidal cycles, because the spatially extensive surge requires longer time to attenuate.

Watch out for undue caution in statements like "flood risk may increase with climate change" - it's pretty unequivocal that sea level rise is happening, will happen, and will increase coastal flood risk.

We have changed the text.

I find the consistent lower response of ERA5 quite surprising, and I'd like to know more about why this is seen, to know more about whether we should be using it elsewhere. In Fig 5, ERA5 appears to be lower than the other models outside of the storm window. What is the alignment like when there is no storm? Does it vary seasonally? It would be good to check in case there is just a constant bias that could be corrected easily.

We have not found a systematic bias in the 10m-height wind speeds, the major differences seem indeed to be limited to extreme wind events. Here we include two examples of wind speed and water level comparison of ERA5 and UERRA for Feb. 1962 and Jan. 1976 storms with an extended period, so that the calm conditions are also visible. The peak differences in wind speed and especially in water level coincide with the storm peak, whereas for moderate and calm conditions the differences are relatively small.
On a large scale, Campos et al. (2022) compared surface winds from ERA5 with satellite data globally. They found a very good agreement of the surface winds, especially for non-extreme conditions. They also pointed out that very strong winds in the North Atlantic were underestimated by ERA5 and the higher the percentiles, the larger was the discrepancy. Unfortunately, major part of the North Sea was omitted from their analysis due to too shallow water and related difficulties with the satellite data. However, these findings are in line with our results.

Campos, R.M.; Gramcianinov, C.B.; de Camargo, R.; da Silva Dias, P.L. Assessment and Calibration of ERA5 Severe Winds in the Atlantic Ocean Using Satellite Data. Remote Sens. 2022, 14, 4918. https://doi.org/10.3390/rs14194918

[Figure]

Wind speed and water level Feb 1962 and Jan 1976

Related, line 420: The study hasn't accounted for mean SLR. Would it be hard to do so, eg add a mean SL estimate to the model data? If not, when in time is it correct - would you expect the model to be too low now or too high at some date in the past?

The SLR in the region manifested itself in changes of about 20 cm during the last century, so a systematic error within 20-30 cm could be expected but was not actually detected, possibly it is masked by a stronger uncertainty source. Additionally, along with the secular sea-level changes there were substantial local changes in the bathymetry due to natural as well as anthropogenic causes. These changes were also not included in the present study. The goal of present work was not to provide the most accurate reconstruction of a particular storm event but rather to assess and compare the abilities of different reanalyses and how the quality of reconstruction depends on the type of storm. So, we decided in favour of consistency rather than accuracy, keeping in mind that the historical bathymetry data are often not available with the desired and unchanging accuracy, especially for earlier events.

If the storm type is important I suggest arranging fig A1 by the 3 clusters.

We think, each storm event has its own individual pressure pattern, so by arranging the storm systems in three cluster, the differences would be lost.

Then also sort by these clusters in other figs (eg Fig 6) so we can compare them more easily.

We have changed the Figure 6 according to your recommendation, see next page

[Figure]

**Figure 6**: Maximum water levels in meter above normal Amsterdam level (Normaal Amsterdams Peil, NAP) for the three selected locations and the eight storm events. The different symbols and colours represent the different atmospheric forcing (see Table 2 and Fig. 4). The black diamonds stand for the observed water levels during high tide (filled) and low tide (unfilled). A red star marks the maximum water level from the tidal shift experiment for a selected member of the 20CRv3 reanalysis. The grey horizontal lines mark the level of very severe storm tide (top), severe storm tide (middle) and storm tide (bottom) for the respective locations (Table 1). The events are sorted according to the storm tracks crossing at 0° longitude and the types are defined crossing at 8° longitude.

Minor points
What is the pink line in Fig A1?
The line marks the 8°E longitude. The storms are divided in Scandinavia, Skagerrak and Jutland types depending on where their tracks cross this line. We also added the explanation to the figures.

Mostly the English is fine but there's some small grammar errors for a copy edit, eg Line 273 & 403, affect. Line 304, line 382

We have corrected the errors.